# Exploring Data Efficiency in Image Restoration: A Gaussian Denoising Case Study

## ABSTRACT

Amidst the prevailing trend of escalating demands for data and computational resources, the efficiency of data utilization emerges as a critical lever for enhancing the performance of deep learning models, especially in the realm of image restoration tasks. This investigation delves into the intricacies of data efficiency in the context of image restoration, with Gaussian image denoising serving as a case study. We postulate a strong correlation between the model's performance and the content information encapsulated in the training images. This hypothesis is rigorously tested through experiments conducted on synthetically blurred datasets. Building on this premise, we delve into the data efficiency within training datasets and introduce an effective and stabilized method for quantifying content information, thereby enabling the ranking of training images based on their influence. Our in-depth analysis sheds light on the impact of various subset selection strategies, informed by this ranking, on model performance. Furthermore, we examine the transferability of these efficient subsets across disparate network architectures. The findings underscore the potential to achieve comparable, if not superior, performance with a fraction of the data—highlighting instances where training IRCNN and Restormer models with only 3.89% and 2.30% of the data resulted in a negligible drop and, in some cases, a slight improvement in PSNR. This investigation offers valuable insights and methodologies to address data efficiency challenges in Gaussian denoising. Similarly, our method yields comparable conclusions in other restoration tasks. We believe this will be beneficial for future research. Codes will be available at [URL].

## 1 INTRODUCTION

Image restoration, a critical domain within low-level vision [15, 19, 38, 45], addresses the reconstruction or enhancement of images degraded by various distortions. It encompasses a spectrum of challenges including image denoising [44, 49–51], JPEG compression artifact removal [8, 10, 24], image deblurring[18, 41], and super-resolution[9, 21]. Among these, image denoising serves as a quintessential example, aiming to recover clean images from their noisy counterparts, and highlights the unique computational demands distinct from broader classification[6, 36] or pattern recognition tasks.

*ACM MM, 2024, Melbourne, Australia*
© 2024 Copyright held by the owner/author(s). Publication rights licensed to ACM.
ACM ISBN 978-x-xxxx-xxxx-x/YY/MM
https://doi.org/10.1145/nnnnnnn.nnnnnnn

**Unpublished working draft. Not for distribution.**

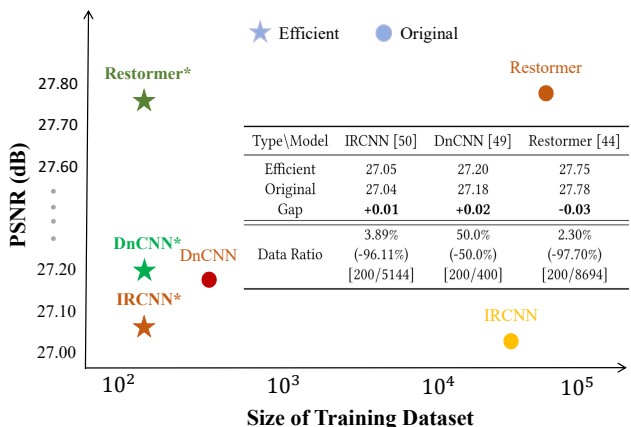

| Type\Model | IRCNN [50] | DnCNN [49] | Restormer [44] |
|---|---|---|---|
| Efficient | 27.05 | 27.20 | 27.75 |
| Original | 27.04 | 27.18 | 27.78 |
| Gap | **+0.01** | **+0.02** | **-0.03** |
| Data Ratio | 3.89%
[200/5144] | 50.0%
(-50.0%)
[200/400] | 2.30%
(-97.70%)
[200/8694] |

**Figure 1: PSNR outcomes for Gaussian image denoising relative to training dataset size at noise level 50 on Set12 [49] testset. We trained representative CNN/transformer models on both the original datasets and efficient subsets with same iterations for fair comparison. Our results underscore the potential to match or surpass fully trained models with significantly fewer training images.**

Recent trends reveal a surge in model complexity [11, 12, 32, 46], particularly with the advent of large-scale vision models [13, 14, 16, 42], necessitating substantial data and computational resources. Although data has consistently remained a pivotal factor influencing deep learning models, the approach adopted by researchers in the domain of low-level vision predominantly falls into one of two categories: either they modify model architectures based on publicly available datasets to achieve state-of-the-art results on popular tasks [22, 37, 44, 48], or they curate bespoke datasets and innovative methodologies for less mainstream tasks [20, 23, 30]. Notwithstanding these developments, there remains a conspicuous gap in the exploration of data itself, particularly in terms of efficiency in the context of image restoration tasks. While some studies like [49, 50] have marginally touched upon the impact of enlarging training data, suggesting only modest enhancements in model performance on certain testsets, a comprehensive exploration into how data influences model learning is markedly absent. This gap underscores the potential for significant advancements in understanding and exploiting data efficiency in this field.

In this paper, we investigates data efficiency in image restoration tasks, using Gaussian image denoising as a case study. Through fundamental hypothesis-driven experiments, we initially substantiate that content information within the images is pivotal for acquiring a perfect denoising model. Our investigation reveals variations in content information complexity across dataset images, highlighting the importance of selecting the most efficient images for training.

Notably, there is currently no concrete method for quantitatively analyzing content information numerically, and removing samples one by one would necessitate $2^n$ model retraining, rendering it impractical. To address these challenges, we employ statistical machine learning techniques to introduce a content information quantification method based on influence function [17]. Specifically, influence function enables the estimation of the impact of training data on test results without the need for retraining the model. We leverage the Hessian-vector product (HVP) method in influence calculation, and propose a stabilized computation process that is particularly suited to image restoration tasks. Through comprehensive experiments, we elucidate the principles of efficient subset selection based on influence scores, assess their applicability across diverse network architectures, and provide an in-depth analysis of model learning behaviors on these subsets. Our findings demonstrate the feasibility of achieving, and in some instances surpassing, the performance of fully trained models with a fraction of the data (see Fig. 1). In summary, the main contributions of this paper are as follows:

- We provide a comprehensive exploration of data efficiency for image restoration research using Gaussian image denoising as a case study, we investigate the effects of content information on model performance, point out that their variations exist inside training datasets and introduce an effective, stabilized method to estimate the influence of each training image.

- We offer an insightful analysis on efficient subset selection based on influence scores, validating their generalization across different network structures, and conducting extensive studies on models' learning behaviors. Our findings reveal that subsets with mid-level influence scores consistently yield optimal test performance.

- Our experimental outcomes underscore the potential to match or exceed the performance of comprehensively trained IR-CNN and Restormer denoising models with merely 3.89% and 2.30% of the original dataset, respectively. To our knowledge, this represents the first in-depth exploration and insightful analysis of data efficiency in the domain of image restoration.

## 2 RELATED WORK

### 2.1 Image Restoration and Gaussian Denoising

Image Restoration, a critical domain within low-level vision which tasked with unraveling the intricacies of elementary image features, such as edges, textures, and optical flow. Image Restoration tasks encompass a wide range of challenges, such as super-resolution [9, 21], denoising [49–51], deblurring [18, 41], JPEG compression artifact removal [8, 10, 24], low-light enhancement [19, 38], and more. In general, the goal is to restore images degraded by specific factors to a higher quality. Gaussian image denoising, a fundamental challenge within the domain of image restoration, has been extensively studied to achieve the highest possible PSNR for images tarnished by noise. Besides, the learning behavior of models in image restoration tasks differs somewhat from high-level vision tasks. High-level vision tasks [25, 53], such as image classification [6, 36], typically involve models that perform stepwise feature extraction on images. These features are then processed through fully connected

layers and softmax functions to output a vector representing the probabilities of belonging to different classes. In contrast, image restoration tasks, like Gaussian image denoising, often require models to establish relationships between current pixels and other pixels. Through these relationships and high-dimensional functions, pixel restoration is achieved. This training process cultivates the model's proficiency in identifying optimal restoration patterns.

Despite the extensive exploration of image restoration techniques, a gap remains in understanding the impact of data on model learning. Specifically, there is a scarcity of research investigating data efficiency in image restoration, indicating a fertile ground for future inquiries.

### 2.2 Data Efficiency

Reducing the size of the dataset while keeping as much information in the remained dataset has long been considered as a challenging problem and a considerable number of researchers have pushed progress in this area theoretically and practically. Data synthesis methods [4, 5, 29, 35, 52] (dataset distillation/condensation) have been the research hotspot recently. For example, Wang *et al.* [35] tried to synthesize a small dataset that can obtain a good performance in the testset by minimizing the classification loss. Another work by Zhao *et al.* [52] inspired by dataset distillation proposed to match the gradients and features to better learn how to synthesize images. Nevertheless, these approaches are computationally expensive and cannot provide a very satisfactory result. Another line of works focuses on data selection/pruning [28, 31, 33, 40, 43], in which methodologies are usually more theoretically explainable. For example, Sorscher *et al.* [33] demonstrated breaking beyond power law scaling by developing a self-supervised pruning metric. Yang *et al.* [43] presented an optimization-based dataset pruning method based on influence function [17] and pruned 40% training examples on the CIFAR-10 dataset with only 1.3% test accuracy decrease.

However, due to the distinct natures and learning patterns of classification tasks and image restoration tasks, it is imperative to rethink how to address data efficiency issues within the sphere of image restoration.

## 3 RETHINKING THE LEARNING PROCESS IN IMAGE RESTORATION

This section delves into the intrinsic learning behavior specific to image restoration tasks. We formulate the general pattern for a learning algorithm to solve a image restoration problem and define that a model's ability to generalize effectively is deeply rooted in the content information—like texture and edges—gleaned from training images. Using the Gaussian image denoising as a case study, we explore how content information influences the development of proficient denoising models uncovering both challenges and insights. Inspired by these observations, we venture into employing the influence function as a novel approach to address these issues.

### 3.1 Formulating Image Restoration Learning

**Notations**. In the realm of image restoration, supervised methods typically leverage a deep neural network $f_\theta$ which maps a

degraded image **y** to an estimated image $f_\theta(\mathbf{y})$, with the clean image **x** as supervision signal. To optimize the parameters of the deep neural network, the target is to minimize the empirical risk function $\frac{1}{n}\sum_{i=1}^{n}\mathcal{L}(x_i, y_i, \theta)$, through solving the following optimization problem:

$$\hat{\theta} \overset{def}{=} \arg\min_{\theta \in \Theta} \frac{1}{n}\sum_{i=1}^{n}\mathcal{L}(x_i, y_i, \theta) \tag{1}$$

**Rethinking Gaussian Image Denoising.** Considering the grayscale Gaussian image denoising as the case, we model the pixel-level distribution of the $i^{th}$ clean image, $x_i$, existing in $\mathbb{R}^{m \times m}$, with $p(x_i)$, assuming an undetermined $m \times m$ dimensional distribution. The additive noise adheres to a Gaussian distribution, denoted as $n \sim \mathcal{N}(0, \sigma^2)$. Consequently, the $i^{th}$ noise-afflicted image is formulated as $y_i = x_i + n$. The pixel-level distribution of $y_i$, $p(y_i)$, integrates the effects of $x_i$ and $n$, expressed as $p(y_i) = p(x_i + n) = p(x_i) + \phi(n)$, where $\phi$ represents the Gaussian distribution's probability density function. A paramount objective for an ideal denoising model is to distinguish and segregate the original image distribution $p(x_i)$ from the noise-induced distribution in $y_i$, irrespective of the underlying pixel distribution in $x_i$. This entails developing a model capable of reversing the noise addition process to isolate $x_i$ from $y_i$, which is a synthesis of $x_i$'s distribution and Gaussian noise. The effectiveness of this separation can be quantitatively measured by a minimal $\epsilon$, representing the discrepancy between the denoised image and the original clean image. The denoising model's success hinges on minimizing $\epsilon$, thereby achieving a near-perfect reconstruction of $x_i$ from $y_i$. The model's theoretical underpinning and empirical validation underscore its potential in advancing denoising methodologies, as outlined below:

$$\text{Given}: x_i, y_i, n \sim \mathcal{N}(0, \sigma^2)$$
$$\text{Perfect Denoising Model}: \hat{x}_i = f(y_i : x_i + n) \tag{2}$$
$$\text{Subject to}: \forall\, x_i \in p(x_i) \ \text{ and } \ ||x_i - \hat{x}_i||_1 < \epsilon$$

**Motivation.** In the pursuit of optimizing image denoising models, **a critical inquiry emerges: the specific contribution of each training image to model performance**. Notably, since Gaussian noise is synthetically introduced using the Gaussian distribution formula, the inherent characteristics of the image, denoted as $x_i$, and its pixel-level distribution, $p(x_i)$, emerge as pivotal factors influencing model learning. Given that $x_i$ can represent diverse contents, including various objects and landscapes, we propose an empirical hypothesis: the content information, particularly edge and texture details, significantly influences the denoising model's theoretical performance ceiling. This perspective underscores the intrinsic link between content information complexity and denoising capability and presented as follow:

**Hypothesis 1.** Given a dataset $\mathcal{D} = \{x_1, ..., x_n\}$ containing $n$ training images, supposing function $C(\cdot)$ summarize content information of a clean image, $\sum_{x_i \in \mathcal{D}} C(x_i)$ represents the whole content information of dataset $\mathcal{D}$, then we hypothesize that the upper-bound performance $\mathcal{P}(\cdot)$ of a DNN model $m$ marked as $\mathcal{P}(m)$ is positively correlated with content information, denoted as $\mathcal{P}(m) \propto \sum_{x_i \in \mathcal{D}} C(x_i)$.

## 3.2 Exploring Effects of Content Information by Blurring the Ground Truth

Blurring images reduces their clarity and informational content [18, 41], posing challenges to human perception and computational models alike. This degradation effect allows us to investigate the role of content information in denoising model performance.

**Definition 1** (*r*-blur dataset). Given a dataset $\mathcal{D}$, we construct an $r$-blur dataset $\hat{\mathcal{D}}_r$ by applying a blurring operation with kernel size $r$ to each image. Training DNN model on $r$-blur dataset $\hat{\mathcal{D}}_r$, we can define that the upper-bound performance $\mathcal{P}(m)$ inversely correlates with $r$ based on Hypothesis 1.

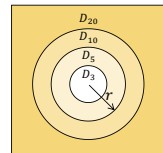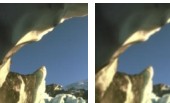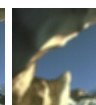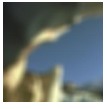

Blur dataset                    Increase blur kernel size to destroy image content

**Figure 2: Synthetic datasets generated by applying image blur functions demonstrate how increased blur kernel size degrades edges and textures, reducing the available content information in ground truth images.**

**Synthetic Dataset Construction.** To study the impact of content information on model learning, we generate four synthetic datasets based on the BSD dataset [3], each with a blur effect at a different level, from small to large. We use the mean filter (see Equation (3)) to blur images and set different kernel sizes to perform local convolutions on pixel regions, resulting in distinct levels of blurriness in the generated datasets (see Fig. 2).

$$I(x, y) = \frac{1}{r \times r}\sum_{i=0}^{r-1}\sum_{j=0}^{r-1} I(x + i - s, y + j - s) \tag{3}$$

$r$ is the kernel size and $s = \frac{r-1}{2}$ presents the center shift. We use $r = \{3, 5, 10, 20\}$ to control blurriness in the generated data. We obtain four synthetic datasets $\mathcal{D}_r = \{\mathcal{D}_3, \mathcal{D}_5, \mathcal{D}_{10}, \mathcal{D}_{20}\}$, and the blurriness increases with the value of kernel size $r$. This aids in comprehending the connection between content information and the upper-bound performance of denoising models.

**Content Information Influences What Model Learns.** We conduct experiments with Definition 1 on the synthetic data to validate Hypothesis 1. We train IRCNN [50] on the synthetic datasets and expect the model's performance on test datasets would inversely correlate with the kernel size $r$, as less blurring datasets carry more distinctive content information than others. To evaluate this, we measure their performance over 1000k iterations training with a fixed learning rate 1e-4, calculating the PSNR/SSIM values on the Set12 [49] testset. Our findings, as detailed in Tab. 1, reveal a substantial decrease in PSNR/SSIM values, particularly for $\mathcal{D}_{20}$ which exhibits the most significant blurring effect, resulting in a 4.81dB decrease.

Remarkably, even the dataset with the least blurring, $\mathcal{D}_3$, exhibited a notable 0.83dB decrease in PSNR. These results underscore the critical role of distinct content information in guiding the learning process of denoising models. Consequently, this leads us to two

pivotal inquiries: **(1) Is content information variability inherent within a single training dataset, and does it differ across samples? (2) How can we harness this attribute to enhance data efficiency?**

**Table 1: Results of Gaussian image denoising by training IRCNN on four synthetic datasets and original dataset. Performance metrics illustrate the decline in PSNR/SSIM with increasing dataset blurriness.**

| Dataset | $\mathcal{D}$ | $\hat{\mathcal{D}}_3$ | $\hat{\mathcal{D}}_5$ | $\hat{\mathcal{D}}_{10}$ | $\hat{\mathcal{D}}_{20}$ |
|---------|------|------|------|------|------|
| PSNR↑ | 27.03 | 26.20 | 25.03 | 23.30 | 22.22 |
| SSIM↑ | 0.7791 | 0.7614 | 0.7250 | 0.6477 | 0.6054 |

## 3.3 Explore Data Efficiency Inside Dataset

So far, our investigation into image restoration and Gaussian image denoising, utilizing blur function to destroy the content of ground truth images, confirms the significant influence of content information on model training outcomes. It suggests that training images vary in content quality, which if properly identified and leveraged, could potentially address data efficiency challenges. However the main obstacles are:

- How to conduct quantitative content information analysis directly on each training image still remains an intractable problem.
- One intuitive method is to evaluate the test performance drop caused by removing each image. However, this is not impractical because it requires re-train the model for $2^n$ times given a dataset with size $n$.

To circumvent these challenges, we propose a methodology for efficiently and quantitatively assessing the impact of individual training images without necessitating model re-training.

## 3.4 Influence Estimation Theory

Considering a denoising problem from noised image space $\mathcal{Y}$ to clean image space $\mathcal{X}$ with Gaussian noise distribution $\mathcal{N}(0, \sigma^2)$. We are given training points $\{z_1, ..., z_n\}$, where $z_i = (x_i, y_i)$. Let us start by studying the model parameter $\hat{\theta}$ change of removing each single training image $z$ from the training dataset $\mathcal{D}$. This change can be formulated as $\hat{\theta}_{-z} - \hat{\theta}$, where $\hat{\theta}_{-z} \overset{\text{def}}{=} \arg\min_{\theta \in \Theta} \frac{1}{n-1} \sum_{z_i \in \mathcal{D}, z_i \neq z} \mathcal{L}(z_i, \theta)$. However, due to $n$ is usually not small and the training denoising model needs a large number of iterations, therefore retraining the model for each removed $z$ is unacceptable and time-consuming.

Fortunately, the research of influence function in machine learning theory [17] provides us with an accurate and fast estimation of parameter change caused by weighting an sample for training. For a training image $z$ and $z$ is weighted by a small $\eta$, the new parameters can be obtained through Equation (4):

$$\hat{\theta}_{\eta,z} \overset{\text{def}}{=} \arg\min_{\theta \in \Theta} \frac{1}{n} \sum_{i=1}^{n} \mathcal{L}(z_i, \theta) + \eta \mathcal{L}(z, \theta) \tag{4}$$

Thus assigning $-\frac{1}{n}$ to $\eta$ is equal as removing the training image $z$ from original training dataset. A classic result [39] tells us that the influence of upweighting $z$ on the parameter $\hat{\theta}$ can be calculated by Equation (5):

$$\mathcal{I}_{\text{up, param}} = \frac{d\hat{\theta}_{\eta,z}}{d\eta}|_{\eta=0} = -H_{\hat{\theta}}^{-1} \nabla_\theta \mathcal{L}(z, \hat{\theta}) \tag{5}$$

where $H_{\hat{\theta}} \overset{def}{=} \frac{1}{n} \sum_{i=1}^{n} \nabla_\theta^2 \mathcal{L}(z_i, \hat{\theta})$ denotes the Hessian and is positive definite by assumption. Then, we can linearly approximate the parameter change due to removing $z$ without retraining the model by computing $\hat{\theta}_{-z} - \hat{\theta} \approx -\frac{1}{n} \mathcal{I}_{\text{up, param}}(z) = \frac{1}{n} H_{\hat{\theta}}^{-1} \nabla_\theta \mathcal{L}(z, \hat{\theta})$. To further evaluate the influence of upweighting $z$ on the loss at a test point $z_{\text{test}}$, a closed-form expression can be obtained via the chain rule, indicated as Equation (6).

$$\mathcal{I}_{\text{up,loss}}(z, z_{\text{test}}) \overset{\text{def}}{=} \frac{d\mathcal{L}(z_{\text{test}}, \hat{\theta}_{\eta,z})}{d\eta}|_{\eta=0}$$

$$= \nabla_\theta \mathcal{L}(z_{\text{test}}, \hat{\theta})^\top \frac{d\hat{\theta}_{\eta,z}}{d\eta}|_{\eta=0} \tag{6}$$

$$= \nabla_\theta \mathcal{L}(z_{\text{test}}, \hat{\theta})^\top \mathcal{I}_{\text{up, param}}$$

$$= -\nabla_\theta \mathcal{L}(z_{\text{test}}, \hat{\theta})^\top H_{\hat{\theta}}^{-1} \nabla_\theta \mathcal{L}(z, \hat{\theta})$$

In this way, we can efficiently approximate the loss change on the test dataset caused by removing $z$ as $-\frac{1}{n \times m} \sum_{j=1}^{m} \mathcal{I}_{\text{up, loss}}(z, z_{\text{test}_j})$.

## 3.5 Influence Calculation

**Hessian-Vector Products.** To calculate $\mathcal{I}_{\text{up,loss}}(z, z_{\text{test}})$, we need to invert $H_{\hat{\theta}} = \frac{1}{n} \sum_{i=1}^{n} \nabla_\theta^2 \mathcal{L}(z_i, \hat{\theta})$, which requires large amount of operations. The inverse of Hessian is well-studied in second-order optimization and can be avoided by using implicit Hessian-vector products (HVPs). First, $s_{\text{test}} \overset{\text{def}}{=} H_{\hat{\theta}}^{-1} \nabla_\theta \mathcal{L}(z_{\text{test}}, \hat{\theta})$ can be efficient approximated via HVPs and then we can compute $\mathcal{I}_{\text{up, loss}}(z, z_{\text{test}}) =$

---

**Algorithm 1:** Pseudo-code, PyTorch-like

```
# y_test and x_test : test sample (noised and clean)
# times: default = len(train_loader)
# scale : default = 25.0
# calculate first order derivative
v = grad (y_test, x_test, model)
s_estimate = copy (v)
# random shuffle
index = random.choice(times)
for i in range (times) :
    # random sampling
    y_i, x_i = train_loader[index[i]]
    # smooth loss to stabilize hvp
    loss = smooth (cal_loss (model (y_i), x_i))
    hv = hvp (loss, params, s_estimate)
    s_estimate = v+ s_estimate - hv/scale
def hvp (y, w, v):
    grads = grad (y, w)
    p = grads · v
    return grad (p, w)
```

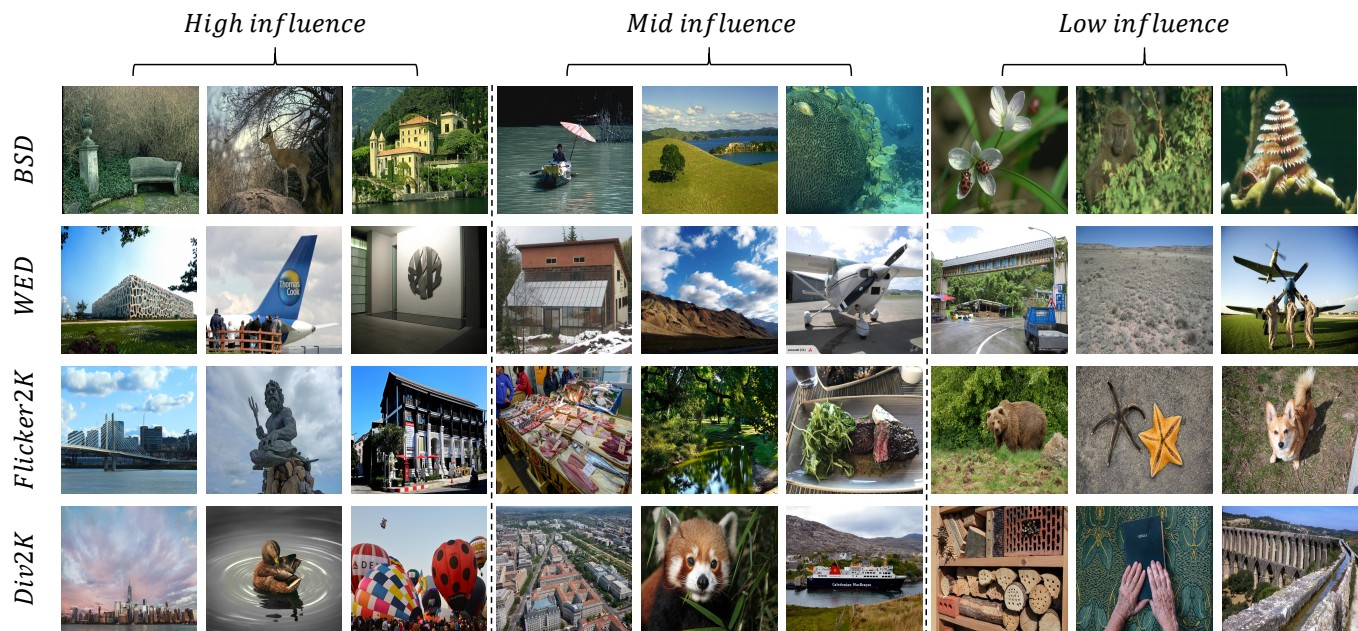

**Figure 3: Showcase of images. We rank images inside each training dataset through influence estimation methodology and create subsets that are directly sampled from three intervals. We define the top K high influence images as the Top subset, mid K influence images as the Mid subset, and lowest K influence images as the Bottom subset, with each subset containing K images.**

$-s_{\text{test}} \cdot \nabla_\theta \mathcal{L}(z, \hat{\theta})$. Because $s_{\text{test}}$ is only related with $H_{\hat{\theta}}^{-1}$ and gradient of $z_{\text{test}}$ with respect to $\hat{\theta}$, so we can efficiently calculate $\mathcal{I}_{\text{up,loss}}(z_i, z_{\text{test}})$ across all training points $z_i$ with only one calculation of $s_{\text{test}}$.

**Stabilized Stochastic Estimation.** We use the stochastic estimation method developed by [1] to get an estimator that only samples a single point per iteration. **Algorithm 1** provides the pseudo-code of stochastic estimation to calculate $s_{\text{test}}$. For each iteration, the HVP method uses two back-prop and one element-wise products to estimate hv (the product of Hessian and $v$), then $s_{\text{estimate}}$ can be calculated by $v + s_{\text{estimate}} + hv/\text{scale}$. In the initial use of the $L_1$ pixel loss (MAE, defined as $||\hat{I}_i - I_i||_1$) for gradient calculation, the problem arises due to the fact that the first-order derivative of the $L_1$ pixel loss is a constant value (if not zero point) that is not dependent on the current input values. Consequently, when this constant value is multiplied and summed with the HVP method and iteration operation, the estimated matrix values might become too large. This, in turn, leads to a numerical explosion during the computation of $s_{\text{estimate}}$. To stabilize the calculation, we originally propose to use the smooth function on the calculation of $L_1$ pixel loss as $\frac{1}{2}||\hat{I}_i - I_i||_2$, if $||\hat{I}_i - I_i||_1 < 1$, as presented in Equation (7):

$$s_{\text{test}} \overset{\text{def}}{=} H_{\hat{\theta}}^{-1} \nabla_\theta \mathcal{L}(z_{\text{test}}, \hat{\theta})$$
$$= \text{hvp (loss, params, s\_estimate)}$$
$$loss = \begin{cases} \frac{1}{2}||\hat{I}_i - I_i||_2 & \text{if } ||\hat{I}_i - I_i||_1 < 1 \\ ||\hat{I}_i - I_i||_1 - 0.5, & \text{otherwise} \end{cases} \tag{7}$$

Given the effective performance of the denoising model, we observe minimal absolute pixel discrepancies between the denoised image $\hat{I}_i$ and the original image $I_i$. This minimal deviation therefore contributes to the stability of the entire stochastic estimation process with our proposed stabilization method.

**Table 2: With the influence scores and selected subsets (Top, Mid, Bottom) from WED, we train IRCNN models on the identified subsets and full dataset. Notice that Mid subset only has a size of 3.89% compared with full dataset but obtains best PSNR.**

| Trainset\Type | Bottom | Top | Mid | Full Size |
|---|---|---|---|---|
| WED | 27.000 | 26.963 | 27.051 | 27.038 |

## 4 EXPERIMENT

### 4.1 Experiment Setup

We perform experiments on grayscale Gaussian image denoising with thorough analysis (Similar conclusions on other restoration tasks can be found in supplementary materials). We assess the data efficiency across four benchmark training datasets: BSD500 [3] (400 images), WED [26] (4744 images), Flickr2K (2650 images) [34], and DIV2K (900 images) [2], referenced respectively. We employ two CNN architectures, IRCNN [50] and DnCNN [49], alongside the powerful vision transformer model Restormer [44], with Set12 [49] and BSD68 [27] testset, to evaluate their performance in our experiments. All experiments are conducted in PyTorch framework with

**Table 3: Summary of grayscale Gaussian image denoising task results using efficient Mid subsets (size 200) from four training datasets. We train IRCNN, DnCNN, and Restormer respectively on each subset (1000k iterations for CNN models and 300k iterations for transformer model), and record their best PSNR results on the Set12 testset. We also train them on their original training datasets with same iterations for fair comparison. IRCNN is trained with BSD and WED (5144 images), while one manually selected training dataset from ImageNet (400 images) is not available. The best results (PSNR) are highlighted in blue.**

| Efficient Subset | Ratio | IRCNN [50] | | | Ratio | DnCNN [49] | | | Ratio | Restormer [44] | | |
|---|---|---|---|---|---|---|---|---|---|---|---|---|
| | | $\sigma = 15$ | $\sigma = 25$ | $\sigma = 50$ | | $\sigma = 15$ | $\sigma = 25$ | $\sigma = 50$ | | $\sigma = 15$ | $\sigma = 25$ | $\sigma = 50$ |
| BSD* | | 32.693 | 30.262 | 27.020 | | 32.858 | 30.442 | 27.177 | | 32.713 | 30.476 | 27.277 |
| WED* | 200 | 32.753 | 30.298 | 27.051 | 200 | 32.908 | 30.444 | 27.198 | 200 | 32.956 | 30.583 | 27.442 |
| Flickr2K* | (3.89%) | 32.685 | 30.259 | 26.992 | (50%) | 32.852 | 30.439 | 27.196 | (2.30%) | 33.079 | 30.822 | 27.723 |
| DIV2K* | | 32.694 | 30.249 | 27.004 | | 32.862 | 30.430 | 27.201 | | 33.133 | 30.822 | 27.746 |
| Original dataset | 5144 | 32.748 | 30.314 | 27.038 | 400 | 32.856 | 30.438 | 27.179 | 8694 | 33.252 | 30.900 | 27.780 |

RTX 4090 GPU. In model training, we use Charbonnier loss [7], Adam optimizer with $\beta_1 = 0.9$ and $\beta_2 = 0.999$. For fair comparison, models are trained over same iterations and learning rate is fixed to $1e^{-4}$ without any modifications, for data augmentation, we use horizontal and vertical flips and obtain random $128 \times 128$ patches.

## 4.2 Main results

### 4.2.1 *Efficient Image Identification via Proxy IRCNN*. The
process of determining the influence of individual images on model parameters typically involves computationally intensive gradient calculations. To streamline this, we employ IRCNN as a proxy model, enabling the identification of subsets of images that exhibit high efficiency. This strategy involves training the proxy IRCNN model on a composite dataset derived from four distinct training sources, with noise levels set at 15, 25, and 50. Such an approach ensures the model captures a comprehensive representation of each image's unique characteristics, thereby yielding more accurate influence scores. Utilizing these scores, we select three efficient subsets: Top, Mid, and Bottom based on their score intervals, as illustrated in Figure 3. To concisely evaluate the effectiveness of our subset selection method, we trained IRCNN models on the identified Top, Mid, and Bottom subsets and compared their PSNR performance against that of an IRCNN model trained on the full dataset (5144 images, WED + BSD, in accordance with its official configuration), over an identical span of 1000k iterations. The outcomes of this comparison are summarized in Tab. 2. This experiment yielded two noteworthy observations:

- As indicated in Tab. 2, **the Mid subset demonstrates better PSNR performance relative to both the Top and Bottom subsets, underscoring the efficacy of Mid subset selection in optimizing PSNR outcomes**.
- Remarkably, training the IRCNN model on merely 200 images deemed efficient (3.89%) not only matches but potentially exceeds the PSNR performance achieved using the entire 5144 image training set (WED + BSD). **This finding suggests that a substantial portion of the data (96.11%) may be superfluous, having a negligible impact on the learning efficacy of the model.**

### 4.2.2 *Generalize to DnCNN and Restormer*. Building upon
the insights from Section 3, the direct correlation between data

efficiency and the intrinsic content information of the datasets has been established. This foundational understanding allows us to extend the training to include other models, such as DnCNN and Restormer, using the efficient subsets identified via the IRCNN proxy model. Specifically, we focus on the Mid subsets (show best performance in Tab. 2) derived from the BSD, WED, Flickr2K, and DIV2K datasets for this phase of experimentation. When training DnCNN and Restormer on these subsets, our results, delineated in Tab. 3, reveal significant findings.

For IRCNN, the Mid subset sourced from WED consistently delivers the highest PSNR across the evaluated subsets, achieving superior performance at noise levels 15 and 50 with only a minimal performance decrement at noise level 25, despite utilizing merely 3.89% of the dataset. This efficiency in data usage while maintaining or enhancing performance metrics underscores the effectiveness of our subset selection methodology.

Cross-model training further validates our approach; the Mid subsets consistently outperform training on the full dataset for the DnCNN model. In the case of Restormer, despite the dataset comprising only 2.30% of the original, the decrease in PSNR is marginal: minimal to 0.03 dB for noise level 50. These results indicate the transformer model's heightened sensitivity to image size, with larger images from Flickr2K and DIV2K contributing to its performance due to increased pixel availability. Conversely, CNN architectures, such as DnCNN, show a preference for efficient subsets from WED, suggesting a nuanced interaction between data efficiency and network architecture. This exploration indicates that *the principle of optimal data efficiency is also intricately linked to the architecture of the network.*

### 4.2.3 *Extensions on Restormer*. Building on the observations
detailed in Tab. 3, where IRCNN and DnCNN models trained on Mid subsets surpassed the performance of models trained with full datasets, we note a slight performance decrement in Restormer under similar conditions. This subsection delves deeper into the Restormer's adaptability and performance nuances.

**Evaluating The Impact of Mid Subset Size Variations.** A key area of inquiry pertains to the potential of achieving or exceeding full dataset performance through adjustments in the size of the Mid subset. This exploration is motivated by the desire to understand the implications of such modifications on model performance. Subsequently, we conducted experiments with Mid subset sizes of K

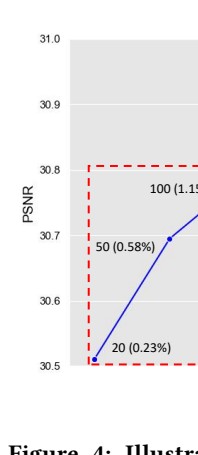

**Figure 4: Illustration of PSNR's progressive change as Restormer models are trained on increasingly larger subsets, presenting the trend of improved model performance with greater subset sizes.**

= 20, 50, 100, 300, specifically chosen from the DIV2K dataset due to its optimal Restormer performance in Tab. 3. Each subset size was assessed under a noise level of 25, adhering to the previously established train/test regimen. The findings, as depicted in Fig. 4, illustrate a pronounced PSNR decline with the smallest subset (20 images, constituting 0.23% of the full dataset), recording a PSNR of 30.511dB—0.389dB below the full dataset benchmark. Expanding the subset to 100 images (1.15%) ameliorated this gap, with a PSNR of 30.767dB, merely 0.133dB shy of the full dataset result. A further increase to 300 images (3.45%) closely matched the full dataset performance, exhibiting a PSNR of 30.832dB, just 0.068dB lower. Although it remains challenging to surpass the performance of full-size dataset, this trend underscores a **diminishing marginal return** on performance gains beyond the 100-image subset size.

**Table 4: Results on testset BSD68 [27]. Only 0.02dB~0.06dB gap in PSNR using 2.3% training data, which is similar with Table 3.**

| Efficient Subset | Ratio | Restormer [44] | | |
|---|---|---|---|---|
| | | $\sigma = 15$ | $\sigma = 25$ | $\sigma = 50$ |
| BSD* | | 31.678 | 29.194 | 26.220 |
| WED* | 200 | 31.734 | 29.279 | 26.326 |
| Flicker2K* | (2.30%) | 31.792 | 29.413 | 26.483 |
| DIV2K* | | 31.849 | 29.411 | 26.501 |
| Original dataset | 8694 | 31.905 | 29.447 | 26.524 |

**Restormer Performance on BSD68 Testset.** To extend our examination of Restormer's capabilities, we also evaluated its performance on the BSD68 [27] testset. As indicated in Tab. 4, the Mid subset derived from DIV2K continued to lead in performance among the four Mid subsets, albeit with a slight decrement (ranging from 0.02dB to 0.06dB) relative to the full dataset—a gap more narrow than that reported in Tab. 3. This parallel in findings further elucidates the nuanced yet consistent behavior of the Restormer model across different testing scenarios, highlighting the efficacy of optimized Mid subsets in approaching, the performance achievable with much larger datasets.

## 4.3 Analysis Experiments

In the preceding section, we have presented the primary experimental findings and highlighted the generalization of Mid subsets across various network architectures. In this section, for a better understanding of model's learning, we present thorough analysis experiments on both Top, Bottom and Mid selection methods. First, we conduct experiments under more constrained data settings, we shrink dataset to a smaller size and compare their performances on Top, Mid, Bottom subsets. Then, we comprehensively analyze model's learning behavior with these three subsets, finding that Mid subset presents both faster convergence speed and better convergence result. Furthermore, we offer visualization of frequency spectrum from three subsets and provide a perspective from frequency and gradient to understand how and why Mid selection outperforms other selection methods. All analysis and subset selections are conducted on original WED [26] with noise level 50 unless mentioned otherwise.

**What If We Further Shrink Three Training Subsets?** It's instructive to explore the difference on model performance if we reduce the selection size of three subsets from K = 200 to a lower number. In this case, we train IRCNN with noise level 50 on less data and record their corresponding best PSNR performances. As shown in Tab. 5, we can discover two phenomenons:

- Mid selection method obtains the best PSNR performance in all experiments.
- The smaller the dataset size is, the larger the Max Diff among the three selection methods becomes, this demonstrates that data efficiency possesses a much a higher status in scenarios lack of training data.

**Table 5: Training IRCNN with noise level 50 on three subsets (Bottom, Mid, Top) selected from WED dataset. We record their best PSNR performances in case of using only 0.39%, 0.97%, 1.94% of training data (5144 images).**

| Size\Type | Bottom | Mid | Top | Max Diff (Mid-Top) |
|---|---|---|---|---|
| 20 (0.39%) | 26.766 | 26.889 | 26.747 | +0.142 |
| 50 (0.97%) | 26.898 | 26.997 | 26.888 | +0.109 |
| 100 (1.94%) | 26.952 | 27.019 | 26.937 | +0.082 |

**What to Learn: Outlier or Basis?** From the definition of influence estimation, it is a little counter-intuitive that the Mid selection method always obtains the best PSNR results. Our methodology tells us, that the Top selection method can select data that have the largest influence on test loss if discarded in model training, while the Bottom selection method presents the opposite characteristic. We record the model performances in whole learning process and visualize the PSNR differences between three subset selection methods (subset size: 20), as presented in Fig. 5. Interestingly enough, We can see that Bottom subset and Mid subset demonstrate similar test performance in the beginning stage, but gradually present larger PSNR gap with more iterations, Top subset seems hard to learn by model so that leads to a max PSNR drop of 0.5 db, but gradually converge to around 0.2 db drop. Similar with [33], from machine learning perspective, Top subset acts like outliers (or hard examples), but Bottom subset acts like basis data (or easy examples),

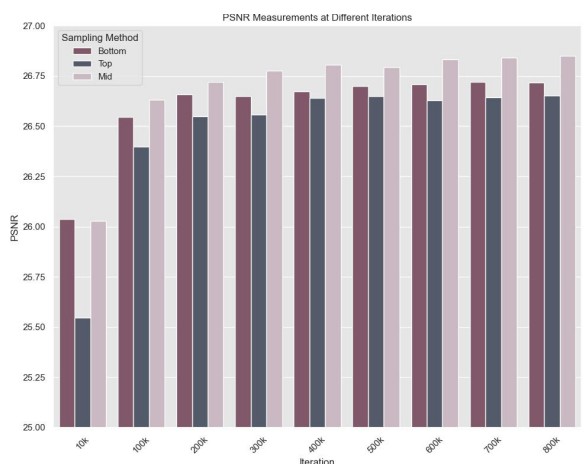

(a) Comparison of PSNR value under different iteration times.

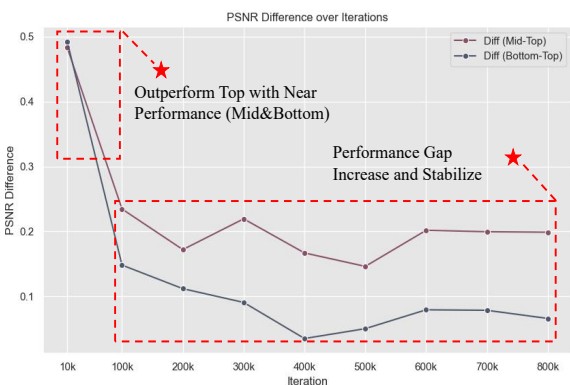

(b) Trend of PSNR difference within three subsets over iterations.

**Figure 5: The learning behaviors of IRCNN model with three subset selection methods (subset size: 20).**

basis data provides coarse-grained information about the target function, while outliers provide fine-grained information. It turns out that neither keeping the hardest samples nor keeping the easiest samples is the best way to obtain the most efficient data in Gaussian image denoising task. Contrary to what might be intuitive, the most effective strategy involves preserving the samples that exhibit smooth characteristics (referred to as the Mid subset).

**Relation to Frequency and Gradient.** In order to gain a deeper understanding of the underlying principles, we select examples from three subsets and apply the Fast Fourier Transform to obtain their corresponding spectrum. As shown in Fig. 6, we can see that high frequency information visually differs in (a)/(b)/(c) which represent examples from Bottom/Mid/Top subsets respectively. Top subset example contains more high frequency information, while Gaussian noise is also high frequency information, thus increase the difficulty for model learning. To obtain a more statistical results, we average all examples with three criteria: 1. gradient based image quality assessment method (IQA [47]) 2. high frequency average pixel intensity (HFAPI, with spectrum center masked). 3. low frequency average pixel intensity (LFAPI, with spectrum center reserved). Tab. 6 presents the quantitative results. It is observed

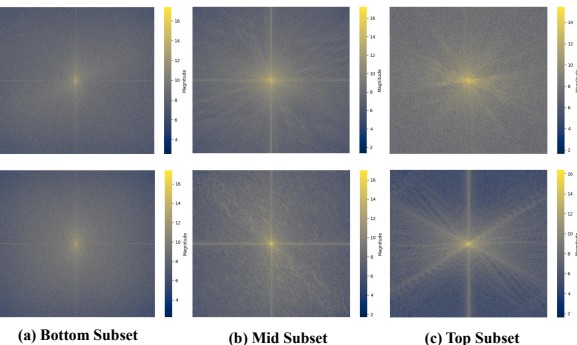

(a) Bottom Subset   (b) Mid Subset   (c) Top Subset

**Figure 6: Spectrum obtained via Fourier Transform, (a)/(b)/(c) column provide examples from Bottom/Mid/Top subsets respectively.**

that the Mid subset exhibits IQA value that is relatively close to the Bottom subset. Simultaneously, the Mid subset contains high-frequency information of almost the same intensity as the Top subset. However, the intensity of low-frequency information in the Mid subset is comparatively lower. Conversely, the Top subset possesses the strongest high-frequency information, but its overall IQA is significantly lower than that of the other two subsets. This observation underscores a limitation in the model's performance on the Top subset. Moreover, the Bottom subset showcases the highest IQA value. This implies that the model is more adept at learning the coarse-grained information provided by the Bottom subset, however, it's difference between the intensity of high-frequency and low-frequency information is minimal among three subsets, which subsequently leads to a lower final result in comparison to the Mid subset.

**Table 6: Statistical results of three criteria on Bottom, Mid and Top subsets (size 20).**

| Type\Criteria | IQA | HFAPI | LFAPI |
|---|---|---|---|
| Bottom | 0.365 | 0.3646 | 0.0279 |
| Mid | 0.3304 | 0.3858 | 0.0242 |
| Top | 0.2692 | 0.3860 | 0.0264 |

## 5   CONCLUSION

In this paper, we navigate the intricacies of data efficiency in the realm of image restoration, with a spotlight on Gaussian image denoising as a pivotal case study. Our comprehensive analysis reveals the profound influence of content information encapsulated within training datasets on the performance of image denoising models. By introducing a novel, stabilized methodology for quantifying this content information, we have enabled a strategic ranking of training images based on their score intervals. Additionally, we perform an in-depth analysis to understand the model's learning behavior under diverse situations. Our experimental results demonstrate that models can achieve, or even exceed, the performance of fully trained counterparts with a fraction of the training data, challenging traditional views on data requirements. We hope our methodological framework and insights will serve as valuable resources for future research.

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
