# OpenReview forum: "Exploring Data Efficiency in Image Restoration: A Gaussian Denoising Case Study"
_acmmm.org/ACMMM/2024/Conference — MM2024 Poster_

### Official Review · Reviewer_Mzmz · 2024-05-22

**Rating:** 3
**Confidence:** 3

**Summary:**

This paper focuses on exploring data efficiency issues specific to low-level image restoration. It uses Gaussian image denoising as a case study in an attempt to dramatically compress data size while maintaining model performance. Specifically, the authors first investigate the effect of the content information contained in the training dataset on the performance of typical image denoising models. The authors then quantify this content information by strategically ranking the training images based on score intervals. In the end, the experimental results demonstrate the feasibility of the proposed method.

**Strengths:**

1. Novelty. Moderate original.
2. Technical Correctness. Probably correct.
3. Reference to Prior Work. References adequate.

**Limitations:**

1. Clarity.

a) The authors have used some overly ornate expressions in this manuscript that prevent the reader from understanding the content of the current paper clearly and easily.

b) The authors should carefully explain the differences between the proposed methods and previous related studies, such as dataset distillation (or dataset condensation), coreset selection, and so on, instead of giving a general conclusion that the performance of these solutions is unsatisfactory.

2. Experimental Validation.

a) As mentioned above, a comparison of different dataset compression methods is required to demonstrate the superiority of the proposed subset selection strategy. This is possible even if the relevant studies are performed in ablation experimental sessions.

b) Considering that the research objective is the image restoration task, some visual comparisons are thus needed to validate the effectiveness of the proposed method, i.e., the ability to maintain a comparable performance to the original model by only training on the selected images. However, these are missing in the current manuscript.

**Suitability:**

2

---

### Official Review · Reviewer_SNZ9 · 2024-05-24

**Rating:** 4
**Confidence:** 2

**Summary:**

This investigation introduces an effective and stabilized method for quantifying content information and delves into the intricacies of data efficiency in the context of image restoration.

**Strengths:**

The paper is sound, clear motivation, well-organized, and articulately presented, ensuring it is easily understandable.

**Limitations:**

1. Data size is key to model performance, especially in the era of large models. In the paper, while training based on multiple datasets and verifying the impact of data effectiveness. (BSD500 (400 images), WED (4744 images), Flickr2K (2650 images) , and DIV2K (900 images)). However, this level of data size is clearly insufficient compared to the data size of today's large models. This is something the author might consider elaborating on.


2. Minor issue: inconsistent formatting of references:
In Proceedings of the IEEE Conference on Computer Vision and Pattern Recognition (CVPR) Workshops.
In 2023 IEEE/CVF Conference on Computer Vision and Pattern Recognition (CVPR).
In Computer Vision – ECCV 2020, Andrea Vedaldi, Horst Bischof, Thomas Brox, and Jan-Michael Frahm (Eds.)

3. Some arXiv references can be replaced with published versions.

**Suitability:**

3

---

### Official Review · Reviewer_UE8S · 2024-05-24

**Rating:** 3
**Confidence:** 4

**Summary:**

The paper provides a thorough investigation into data efficiency in image restoration research, using Gaussian image denoising as a case study. This paper explore the impact of content information on model performance and introduce a stabilized method for estimating the influence of each training image. This paper offers insightful analysis on efficient subset selection based on influence scores, demonstrating their generalization across different network structures and conducting extensive studies on models' learning behaviors. The experimental results highlight the potential to achieve or surpass the performance of comprehensively trained models with significantly reduced datasets.

**Strengths:**

1. This paper offers a comprehensive examination of data efficiency in image restoration, providing valuable guidance for further optimizing training data and strategies.
2. The analysis of learning behaviors from a data perspective contributes to a deeper understanding of neural network learning mechanisms.
3. The effective and stabilized method for quantifying content information appears to be highly practical.

Benifiting from my extensive industrial experience, I understand and endorse the majority of the discoveries presented in this paper. The discoveries described align with findings from numerous industrial practitioners over the years, which deserve broader recognition.

**Limitations:**

As stated above, I can promise that the majority of the discoveries presented in this paper have been confirmed to be true in the industrial field. Unfortunately, I have also found many issues with this paper, especially when I consider it as an academic paper.

1. In L105, the paper claims that *"a comprehensive exploration into how data influences model learning is markedly absent,"* which is actually incorrect. A series of papers closely related to raw image processing, such as those on noise modeling, have long focused on *"how data influences model learning."* I do not find any keywords related to *"raw"* throughout the paper, indicating that this paper significantly lack of research in this area.
I recommend two directly relevant raw denoising works here:
    a) SFRN (*Rethinking Noise Synthesis and Modeling in Raw Denoising*), which provides superior experiments in Sec. 3.2.
    b) PMN (*Learnability Enhancement for Low-Light Raw Image Denoising: A Data Perspective*), its diversity analysis of datasets and data mapping learnability analysis are highly relevant to this paper.
Many conclusions presented in this paper can be pieced together from various papers in the field of degration modeling.

2. The paper focuses on analysis, and the proposed methods are merely minor improvements based on influence estimation theory, lacking novelty from an academic perspective. Furthermore, selecting mid-influence images as a subset is an overly tricky approach, making the criterion unstable. For instance, if I extract low-influence images from a massive dataset as a new dataset, the meaning of mid influence in the new dataset would obviously differ from that in the original dataset.

3. The paper's title risks overclaiming, as conclusions drawn from image denoising may not necessarily generalize to image restoration. Image denoising heavily relies on noise priors, which may differ from tasks relying more on image priors (such as super-resolution). At least, the compressibility of the dataset should be heavily discounted.

4. The writing style of the paper adds to the difficulty of reading, which I would describe as "loan-style introduction." I often encounter confusing descriptions with explanations that are far from the description (position), greatly disrupting the reading experience.

### Additional Comment

The paper raises a very good question, and I love it. It initially impressed me, and I expected to judge it as "Weak Accept" at first. However, unfortunately, it did not provide much incremental information for me, and there are many problems. I believe this question should be addressed in a better way, and I look forward to seeing a revised version of the paper in the future.

**Suitability:**

3

---

### Official Review · Reviewer_kt8g · 2024-05-25

**Rating:** 5
**Confidence:** 3

**Summary:**

The paper explores data efficiency in image restoration, focusing on Gaussian image denoising. It emphasizes the significant impact of content information, such as edges and textures, on model performance. By ranking training images based on their influence scores, the paper identifies efficient subsets for training. Experimental results demonstrate that models can achieve comparable or superior performance using a fraction of the original data. The study provides valuable insights and methods for addressing data efficiency challenges in image restoration tasks.

**Strengths:**

1. This paper introduces a novel and stabilized method to quantify content information in training images, enabling the ranking of images based on their influence.

2. The paper presents comprehensive experiments to validate the hypothesis, explore efficient subset selection strategies, and assess the transferability of efficient subsets across different network architectures.

3. The paper provides an in-depth analysis of model learning behaviors on efficient subsets, shedding light on why mid-level influence subsets consistently yield optimal test performance.

4. The proposed method shows generalization across different network architectures and datasets, indicating its potential applicability to a wide range of image restoration tasks.

**Limitations:**

Some points need to be clarified:

1. How to define the "empirical risk function" in  Equation (1)?

2. Since exploring the effects of content by blurring the ground truth, what is the proposed method's effect in the debluring task?

3. The data selection methods, as a data pre-processing manner, should discuss the calculation load for a fair comparison with those network design manners.

4. Some mistakes in Figure 3 texts.

**Suitability:**

3

---

### Meta-Review · Area_Chair_ZFJW · 2024-07-04

**Recommendation:** Accept (Poster)
**Confidence:** 5

**Metareview:**

The authors provided a response to the reviewers' expressed concerns.

Reviewer Mzmz finds it unsatisfactory as he still has concerns with regard to clarity and experimental validation.

The other three reviewers recommend acceptance, weighing in the ideas, the solution, and the quantitative results.

After carefully checking the paper, the reviews, the rebuttal, and the post-rebuttal justifications, the Meta-Reviewer agrees with the reviewers that paper is "moderately original" (Reviewer Mzmz), presents "comprehensive experiments" and "in-depth analysis" (Reviewer kt8g,Reviewer UE8S), "it is easily understandable" (Reviewer SNZ9), and has practical value (Reviewer kt8g,Reviewer UE8S).

The strengths outweigh the weaknesses and the Meta-Reviewer agrees with the majority of the reviewers that this work makes significant contributions of interest to the community and invites the authors to further improve their work and integrate part of the contents from their response and address the remaining issues in the camera ready paper.